# Dietary Counseling Outcomes in Patients with Lung Cancer in an Upper-Middle-Income Country: An Open-Label Randomized Controlled Trial

**DOI:** 10.3390/jcm13175236

**Published:** 2024-09-04

**Authors:** Busyamas Chewaskulyong, Haritchanan Malairungsakul, Supawan Buranapin, Panas Jesadaporn, Thanika Ketpueak, Thatthan Suksombooncharoen, Chaiyut Charoentum

**Affiliations:** 1Division of Medical Oncology, Department of Internal Medicine, Faculty of Medicine, Chiang Mai University, Chiang Mai 50200, Thailand; thanika.k@cmu.ac.th (T.K.); thatthan.s@cmu.ac.th (T.S.); chaiyut.charoentum@cmu.ac.th (C.C.); 2Division of Gastroenterology, Department of Internal Medicine, Faculty of Medicine, Chiang Mai University, Chiang Mai 50200, Thailand; haritchm@gmail.com; 3Division of Endocrinology and Metabolism, Department of Internal Medicine, Faculty of Medicine, Chiang Mai University, Chiang Mai 50200, Thailand; supawan.b@cmu.ac.th; 4Division of Geriatric Medicine, Department of Internal Medicine, Faculty of Medicine, Chiang Mai University, Chiang Mai 50200, Thailand; panas.j@cmu.ac.th

**Keywords:** lung cancer, dietary counseling, malnutrition, nutritional status

## Abstract

**Background:** Malnutrition harms treatment outcomes, QoL, and survival in lung cancer patients. Effective dietary counseling can improve nutrition, but few randomized controlled trials have focused on lung cancer patients. The objective of this study was to determine if dietary counseling improves nutritional and treatment outcomes when compared to routine care. **Methods**: This open-label parallel RCT was conducted at Maharaj Nakorn Chiang Mai Hospital in Thailand. The investigators used computer-generated blocked randomization to assign patients to dietary counseling by a nutritionist or routine care. The nutritionist sessions occurred before treatment, with follow-ups at 3–4 weeks and 12 weeks. The primary outcome was the mean percentage change in the body weight of patients at 12 weeks. Secondary outcomes included changes in the BMI, nutrition score, QoL, serum albumin level, lymphocyte count, energy and protein intake, treatment response, PFS, and OS. **Results**: Between April 2020 and May 2022, after completing recruitment, 80 lung cancer patients were randomized: 43 to dietary counseling and 37 to routine care. The dietary counseling group showed significant benefits, with smaller decreases in body weight at 3–4 weeks (−0.8% vs. −2.6%, *p* = 0.05) and 12 weeks (−1.1% vs. −4.3%, *p* = 0.05). They also had higher energy and protein intake levels and better treatment response rates. The secondary outcomes and significant adverse events did not differ significantly between the groups. **Conclusions**: Dietary counseling helps to maintain body weight, maintain dietary intake, and enhance treatment responses in lung cancer patients. Although not all nutritional markers or survival outcomes were affected, these findings highlight the importance of early nutritional interventions.

## 1. Introduction

The prevalence of malnutrition among patients with cancer ranges from 20% to 70%, with the severity contingent on the tumor type, staging, clinical setting, and treatment modalities, including both locoregional and systemic treatments [1,2,3]. At the initial medical oncology consultation for solid cancer diagnosis and treatment-naïve patients, 51% exhibited nutritional impairment, 9% were overtly malnourished, and 43% were at risk of malnutrition [1]. Malnutrition severity is positively correlated with the cancer stage [1]. The tumor subsite is a significant determinant of malnutrition, with the highest prevalence observed in pancreatic, esophageal, other gastroenteric, head and neck, and lung cancers [1]. Moreover, advanced cancer stages are associated with a higher risk of malnutrition due to the relationship between tumor burden, inflammatory status, reduced caloric intake, and malabsorption [1]. Specifically, patients with lung cancer have a 35–70% risk of malnutrition [2]. An observational cohort pilot study in advanced-stage non-small cell lung cancer (NSCLC) revealed that 78% of patients required specialist nutritional advice, with 52% in critical need of dietetic input and symptom management [4]. The most commonly reported symptom for patients with advanced NSCLC was a lack of appetite (43%); other symptoms included fatigue (30%), early satiety (26%), and constipation (24%) [4]. Another study reported a high prevalence of symptoms in advanced NSCLC patients, including fatigue (100%), loss of appetite (97%), and shortness of breath (95%) [5]. The concordance between physician- and patient-reported symptoms was lowest for the loss of appetite [5].

Cancer-associated cachexia significantly and independently affects the body’s composition and body weight, which influences clinical outcomes such as lung cancer-specific survival (LCSS) and overall survival (OS) in NSCLC [6]. A meta-analysis found that nutritional status correlates with outcomes such as OS, time to tumor progression, and quality of life (QoL) in patients with lung cancer [2]. Malnutrition is associated with poorer outcomes, including reduced OS, a faster time to tumor progression, and a lower QoL [2]. However, nutritional screening for at-risk patients is not routinely conducted in many hospitals, indicating the ongoing challenges in the management of malnutrition. This indicates that malnutrition problems in hospitals persists. Consequently, malnourished patients are at high risk of developing complications during treatment which can affect their treatment outcomes [7]. A study shows that inadequate pre-treatment nutritional assessments are linked to increased post-treatment complications in patients with stage I-III NSCLC [8]. A systematic review underscored the importance of nutritional interventions in patients with cancer undergoing chemotherapy [9]. Nutritional counseling has been demonstrated to improve QoL, enhance responses to therapy, and increase survival rates [9]. 

The European Society for Parenteral and Enteral Nutrition (ESPEN) [10], the American Society for Parenteral and Enteral Nutrition (ASPEN) [11], and the other national and international organizations have established evidence-based guidelines for nutritional screening and assessment [12]. These guidelines recommend initiating nutritional evaluation during cancer diagnosis and the regular monitoring of nutritional intake, weight changes, and body mass index (BMI). Moreover, they emphasize tailored nutritional interventions, particularly for preventing weight loss in patients with cancer and ensuring adequate nutritional intake [10,11,12]. Nevertheless, malnutrition is often under-recognized in clinical practice, highlighting a critical need for early and comprehensive nutritional assessment in patients scheduled for a definitive cancer treatment. Therefore, this study aimed to evaluate the baseline nutritional status of patients newly diagnosed with lung cancer, and to assess the impact of dietary counseling on nutritional outcomes in patients undergoing definitive cancer therapies. The goal was to determine whether dietary counseling provides better nutritional, and treatment outcomes compared to routine care.

## 2. Materials and Methods

### 2.1. Study Design

A single-center, open-label, parallel, prospective, randomized controlled trial was conducted in patients with newly diagnosed lung cancer undergoing definitive treatment (chemotherapy, radiotherapy, targeted therapy, and immunotherapy) at the Maharaj Nakorn Chiang Mai Hospital in Chiang Mai, Thailand between April 2020 and May 2022.

### 2.2. Participants

#### 2.2.1. Eligibility and Exclusion Criteria

Patients were eligible to participate if they were aged >18 years, had any stage of lung cancer, underwent definitive treatment (including chemotherapy, radiotherapy, targeted therapy, or immunotherapy), and had an Eastern Cooperative Oncology Group (ECOG) performance status of 2 or lower. Patients with two primary tumors, heart failure, edema or ascites, dysphagia, bowel obstruction, a history of gastrointestinal tract surgery, a life expectancy of less than 1 month, or on total parenteral nutrition were excluded.

#### 2.2.2. Withdrawal Criteria

Patients were withdrawn from this study if they died from a cause other than lung cancer or if they withdrew their consent.

### 2.3. Assessment

#### 2.3.1. Patient Selection, Randomization, and Trial Endpoint

The investigators prescreened lung cancer patients who attended the oncology clinic. Participant eligibility was confirmed based on the inclusion and exclusion criteria, and informed consent was obtained. Subsequently, they were randomly assigned to either the dietary counselling or routine care group at a 1:1 allocation ratio by the investigators. All randomization allocations were performed externally using computer-generated random number codes via blocked randomization (with a block size of four) which available from https://sealedenvelope.com/simple-randomiser/v1/lists (accessed date 27 July 2024). The sequence was not concealed and not blinded until the interventions were assigned. The investigators conducted the recruitment, data collection, and data analysis. Participants randomized to the nutrition counseling group were referred by investigators to a dietitian who provided the nutrition counseling. Those in the routine care group did not receive counseling from a dietitian but continued with their usual nutritional advice practices. The trial would end after 80 patients had been randomized or if the investigator or ethics committee expressed concerns regarding safety, lack of efficacy, or significant new evidence or upon request by the ethics committee. No interim analysis was planned, and there were no stopping guidelines except at the request of the ethics committee or the decision of the investigator. Recruitment began in April 2020, and follow-up continued until the data cut-off date of 20 September 2023.

#### 2.3.2. Patient Assessment

Patients were assessed at baseline before treatment, and during follow-up visits at 3–4 weeks and 12 weeks at the oncology clinic. Demographic data (e.g., age, gender, comorbid diseases, and smoking history), lung cancer staging (AJCC 8th edition), tumor biomarkers, definitive treatment (i.e., chemotherapy, radiotherapy, targeted therapy, immunotherapy, and combination therapy), ECOG performance status, QoL using the Thai Modified Functional Living Index Cancer Questionnaire version 2 (T-FLIC 2), and nutritional data (e.g., body weight, weight loss percentage, height, BMI, grip strength and NT 2013 score [13], which is a recommended nutrition screening and assessment tool for practical clinical use in Thailand, and energy and protein intake via food diaries) were collected at baseline as well as at 3–4 weeks, and 12 weeks during follow-up visits. Survival data, including progression-free survival (PFS), defined as the time from randomization to documented disease progression or death from any cause, whichever occurred first, and OS, defined as the time from randomization to death from any cause, were recorded. Laboratory data (e.g., complete blood counts, serum albumin levels, and absolute lymphocyte count), grip strength, and gait speed were also collected at baseline, at 3–4 weeks, and at 12 weeks. Adverse events were record as the routine practice.

#### 2.3.3. Tumor Assessments

Tumor assessments of the chest and upper abdomen were performed using computed tomography (CT) as part of the routine schedule for evaluating overall responses. These assessments were conducted before treatment and after 4 or 6 cycles, or in cases of clinically suspected progression. Brain imaging was performed based on individual patient needs, particularly for those with brain metastases or clinically suspected brain involvement. Responses were evaluated according to RECIST version 1.1.

### 2.4. Nutritional Intervention

Counseling was conducted before starting treatment based on the ESPEN guidelines (2016/19) and considering individual comorbidities such as diabetes, CKD, and others. These guidelines recommend an energy requirement of 25–30 kcal/kg per day and a protein requirement of 1–1.5 g/kg per day. For patients unable to meet their nutritional needs through normal food intake, oral nutritional supplements (ONSs) were recommended. Dietary counseling in the intervention group was individualized and rigorous, with the dietitian focusing on maintaining and improving patients’ body weight as well as their energy and protein intake. The dietitian evaluated the nutrition scores and details of energy and protein intake through interviews and a food diary. Nutritional counseling was given individually to the intervention group during the first visit. Nutritional counseling was intensively reinforced during follow-up visits at 3–4 weeks and 12 weeks to emphasize nutritional education and to evaluate daily energy and protein intake using a diet diary.

The routine care group received general dietary recommendations from a physician as a part of their routine care before undergoing treatment, without advice from a dietitian. In both groups, patients were provided with a food diary to record the type, estimated intake, and frequency of meals, as well as the amount and type of ONSs consumed over a 24 h period, to evaluate their daily energy and protein intake. The patients were instructed to update their diet diary at least 3–4 times per week. Both groups were prescribed regular food adjusted to their individual daily diets, with supplementary diets or vitamins allowed if necessary.

### 2.5. Study Objectives

The objectives of this study were to evaluate the impact of dietary counseling on nutritional outcomes in patients undergoing definitive cancer therapies, with the following specific outcomes in focus:

The primary outcome was the mean percentage change in the body weight of patients at 12 weeks after first-line treatment (i.e., chemotherapy, radiotherapy, targeted therapy, or immunotherapy). This reflects nutritional status following nutritional counseling and will impact clinical outcomes. Secondary outcomes included the mean percentage change in BMI, nutrition score (NT 2013), QoL using the T-FLIC 2, serum albumin levels, absolute lymphocyte count, and energy and protein intake, which reflects the assessed differences between baseline and 3–4 weeks, and between baseline and 12 weeks. Secondary outcomes also included the overall response rate (ORR), and progression-free survival (PFS). Overall survival (OS) became an exploratory outcome after the trial commenced due to its impact on the overall status of the patients.

### 2.6. Statistical Analyses

Based on a previous study conducted in Maharaj Nakorn Chiang Mai Hospital [14], a sample size calculation was performed to detect the minimum relevant differences in the mean percentage change in the body weight of patients at 12 weeks between groups with a 95% confidence interval and 80% test power, with alpha = 0.05 (using a two-sample *t*-test). These calculations indicated that 39 patients were required per group. To allow for a 10% dropout rate, we adjusted the number to 45 patients per group. It turned out to be around 40 patients for each group (80 patients in total for both groups).

Overall analyses were conducted in the intention-to-treat population. Descriptive data are presented as the mean and standard deviation (SD) or median (p25, p75) as appropriate. The categorical variables are presented as numbers and percentages.

Continuous variables between the two groups were compared using the Student’s *t*-test or the Mann–Whitney U test, while categorical variables were compared using the chi-squared test or Fisher’s exact test as appropriate. According to repeated measurements of the outcome data (i.e., BW, BMI, NT2013 score, QOL, albumin level, absolute lymphocyte count, and energy and protein intake), percentage changes from baseline were calculated using the outcome measurement at 3–4 weeks–baseline and 12 weeks–baseline. With these percentage changes from baseline, within subject variation was adjusted. The Kaplan–Meier method was used to estimate the PFS and OS, while comparisons between groups were performed using the log-rank test. All analyses were performed using Stata software (version 14.0; StataCorp LP, College Station, RX, USA). A two-tailed test with *p* < 0.05 was considered to be statistically significant. The data cut-off date was 20 September 2023.

## 3. Results

### Study Population

Between April 2020 and May 2022, after completing recruitment, 80 newly diagnosed lung cancer patients were enrolled in this study. Among them, 43 and 37 patients were randomly assigned to the dietary counseling and routine care groups, respectively. At the cutoff time for survival analyzed, only 12 and 11 patients remained in the nutritional counseling and routine care groups, respectively. The CONSORT diagram is shown in Figure 1.

The baseline patient characteristics and nutritional status are provided in Table 1 and Table 2, respectively. These attributes were well-balanced between the two groups, except for the QoL and energy intake, which were notably higher in the nutrition counseling group, and there was a lower percentage of lung metastasis in the nutrition counseling group. First-line treatment showed no significant differences between the groups. Regarding baseline nutritional status, the prevalence of underweight (BMI < 18.5 kg/m^2^) patients was 21.25%, and 27.5% of patients experienced weight loss of >5% within the first month, with similar rates between the two groups. The mean body weight was 54.88 ± 10.29 kg, and nearly half of the patients had a normal BMI. Both groups were well-balanced in terms of baseline nutritional status, except that the nutrition counseling group had a better QoL score and a higher energy intake (Table 2).

## 4. Nutritional Assessments

### 4.1. Body Weight

At the 3–4- and 12-week follow-ups, the mean body weight in the nutrition counseling group was similar to that of the routine care group (Table 3). However, the mean percentage of body weight loss was significantly lower in the nutrition counseling group than in the routine care group at 3–4 weeks of follow-up. At the 12-week follow-up, the mean percentage of body weight loss in the nutrition counseling group remained lower than that of the routine care group, although the difference was not statistically significant (Table 4 and Figure 2).

### 4.2. Body Mass Index (BMI)

At the follow-up times of 3–4 and 12 weeks, the mean percentage loss in BMI in the nutrition counseling group was lower than that of the routine care group (Table 3). However, these differences were not significant (Table 4 and Appendix A). 

### 4.3. Nutrition Score

At the first follow-up at 3–4 weeks, a lower nutritional score was observed in both groups, indicating an improvement in nutritional status, with lower scores indicating better nutrition. At 12 weeks, the nutritional scores were lower than those at 3–4 weeks. The mean nutrition score and mean percentage change were better in the nutrition counseling group than in the routine care group at both the 3–4- and 12-week follow-up visits, although these differences were not statistically significant (Table 3 and Table 4 and Appendix A).

### 4.4. Quality of Life

The mean QoL dimension scores for the nutrition counseling group were higher at baseline, indicating a better QoL than for the routine care group. At the 3–4- and 12-week follow-up visits, no significant difference was seen between the groups; however, the percentage change in QoL improved less in the nutrition counseling group compared to the routine care group. The smaller effect size in the nutrition counseling group may have been a result of their higher QoL scores at baseline (Table 4 and Appendix A).

### 4.5. Grip Strength and Gait Speed

There was no difference in the percentage change of grip strength between the nutrition counseling and routine care groups at the 3–4-week follow-up. The data showed an improved percentage change of grip strength at the 12-week follow-up in the nutrition counseling group, with a significant percentage change in grip strength at the 12-week mark (Table 3 and Table 4). For gait speed, the results were similar between the nutrition counseling and routine care groups at the 3–4-week follow-up. At the 12-week follow-up, the reduction in gait speed was lower in the nutrition counseling group than in the routine care group, and the percentage change in gait speed reduction was also lower in the nutrition counseling group, although these differences were not statistically significant compared to the routine care group (Table 3 and Table 4, and Appendix A).

### 4.6. Serum Albumin

No significant difference was observed in the serum albumin levels between the two groups at the 3–4-week follow-up. By the 12-week follow-up, the serum albumin levels were higher in the nutrition counseling group than in the routine care group, although the difference was not statistically significant (see Table 3 and Table 4, and Appendix A).

### 4.7. Absolute Lymphocyte Count

No significant differences were observed in the mean and percentage change in the absolute lymphocyte count between the two groups at both the 3–4- and 12-week follow-up times (Table 3 and Table 4, and Appendix A).

### 4.8. Energy and Protein Intake

At baseline, the nutrition counseling group had a higher energy intake than the routine care group, and this trend continued during both the first and second follow-up periods. The percentage change of energy intake was higher in the nutrition counseling group; however, no statistically significant difference was observed (Table 3 and Table 4). There was no difference in protein intake between the nutrition counseling and routine care groups at baseline or at the 3–4-week follow-up. At the 12-week follow-up time, the nutrition counseling group had a significantly higher mean protein intake, although no significant difference was observed in the percentage change in protein intake (Table 3 and Table 4, and Appendix A).

## 5. Treatment Outcomes

A significantly higher best response rate was observed in the nutrition counseling group, with all responders showing partial responses (Table 5). No significant differences were observed in the PFS and OS between the two groups (Table 5, and Appendix A). The subsequent treatments and number of treatment lines were similar between the groups (Table 6).

No intervention-related serious adverse events were reported, and no patients from either the intervention or routine care groups reported problems. A significant serious adverse event refers to an adverse event that leads to death, hospitalization, disability or permanent damage, or a congenital anomaly/birth defect.

## 6. Discussion

Malnutrition, weight loss, and low muscle mass in patients with lung cancer are associated with treatment intolerance, leading to worse treatment outcomes, a shorter PFS, worse survival rates, and a lower quality of life [15,16,17]. A retrospective study of patients with lung cancer treated with immunotherapy indicated that severe malnutrition was associated with lower treatment efficacy in a univariate analysis, although this association was not observed in a multivariate analysis [18]. A recent cross-sectional study in Norway revealed that 55.4% of cancer patients had not discussed dietary changes with any healthcare professionals [19].

### 6.1. Main Findings

The findings of this study suggest that invasive individualized dietary counseling for patients newly diagnosed with lung cancer who are receiving specific treatments influenced the percentage changes in body weight and grip strength, indicating trends toward improvements in BMI, NT 2013 score (i.e., nutritional status), serum albumin level, and QoL when compared to the routine care group. Although energy and protein intakes significantly improved in the dietary counseling group, these changes were not statistically significant. The nutrition counseling group also demonstrated benefits in terms of the best response. However, there were no differences in the PFS and OS between the two groups.

### 6.2. Nutritional Status

Our study demonstrated that nutrition counseling significantly reduced the mean percentage of weight loss at the 3–4-week mark and showed a trend towards reduction at the 12-week mark. These findings are consistent with those of several randomized controlled trials (RCTs) in previous studies. For example, patients with head and neck cancer undergoing antineoplastic therapy experienced less weight loss because of nutritional counseling [20]. Similar outcomes were observed in head and neck cancer patients who underwent radical treatment, such as chemoradiotherapy or surgery with postoperative radiotherapy, and received nutritional counseling combined with oral nutritional supplements [21]. An RCT focusing on individual nutritional counseling in patients with esophageal, gastric, and gynecological cancers also reported improved weight maintenance [22]. RCTs involving various types of tumors have consistently demonstrated improvements in body weight [14]. A pilot RCT in patients with lung cancer undergoing radiotherapy also reported favorable results for weight maintenance, despite the small sample size [23]. A systematic review by Payne et al. indicated that nutritional interventions may benefit patients with advanced NSCLC by improving unintentional weight loss, physical strength, and functional performance [24]. Similarly, evidence-based practice guidelines in 2013 found that dietary counseling and/or oral supplements enhanced dietary intake and weight in patients undergoing chemotherapy, which is consistent with our study’s findings [25].

In contrast, another systematic review which included lung cancer among other types, did not find a significant increase in weight in the dietary counseling group compared to the control group [26]. This discrepancy may be attributed to the variations in cancer types that influenced the study outcomes. Other systematic reviews and meta-analyses also indicated no significant improvement in body weight after nutritional counseling among patients with incurable cancers [27].

To improve short- and long-term weight maintenance, more frequent nutritional counseling and supplementation may be necessary. A systematic review and meta-analysis [28] suggested that oral nutritional supplementation may increase body weight in patients with gastrointestinal, head and neck, and lung cancers undergoing chemotherapy. This benefit is particularly notable in populations at high risk of malnutrition, including the elderly, those with a low baseline body weight, women, and non-Asian patients. Additionally, oral nutritional supplementation has been associated with improvements in patient-generated subjective global assessment (PG-SGA) scores and significant enhancements in QoL [28]. Another systemic review by Planski et al. demonstrated that supplementation with essential nutrients and antioxidants may have a beneficial effect on lung cancer treatment [29].

Regarding other aspects of nutritional status, our study found no significant differences in the BMI, nutritional score, serum albumin level, or absolute lymphocyte count between the two groups. This contrasts with the findings of Sukaraphat et al., who reported improvements in the BMI and nutritional score, but not in the absolute lymphocyte count and serum albumin level, which is consistent with our results [14]. An RCT focusing on patients undergoing chemotherapy and/or radiotherapy demonstrated that dietary counseling following the nutritional care process pathway significantly improved nutritional status compared to standard practices in patients with cancer undergoing treatment [30].

Moreover, systematic reviews have indicated that nutritional counseling provided by registered dietitians leads to an improved nutritional status in patients with head and neck cancer undergoing radiation therapy with or without chemotherapy [31,32]. Regarding grip strength and gait speed, our study evaluated a small number of patients, demonstrating a significant change in grip strength but no difference in gait speed. These findings should be interpreted with caution and warrant validation in a larger sample.

Our study demonstrated an increase in mean energy and protein intake. These findings are consistent with an RCT conducted in gastrointestinal and gynecological cancer patients [22]. Additionally, a systematic review of patients with lung cancer and other types of cancer undergoing chemotherapy and/or radiotherapy indicated that nutritional counseling led to improvements in energy and protein intake [26]. Another systematic review and meta-analysis confirmed that increased energy and protein intakes are achievable. The evidence demonstrates that nutritional counseling significantly improves energy and protein intake in patients with incurable cancers, including lung cancer [27]. However, an RCT involving patients with mixed cancer types reported no differences in energy intake [14].

### 6.3. Treatment Outcomes and Survival

Our study demonstrated a significant increase in the best-response rate in the nutrition counseling group. This finding is supported by an RCT on breast cancer, which showed that diet and exercise interventions were associated with a higher pathological response [32]. However, specific evidence confirming a response improvement in lung cancer patients remains lacking. Data from a systematic review [26], which included an RCT on patients with lung cancer and other cancer types undergoing chemotherapy and/or radiotherapy, did not find any benefits for nutritional counseling in treatment response or improvement of the survival rates. These outcomes are consistent with those of previous studies [27,33,34,35]. Rothpletz-Puglia [36] recently provided a theoretical explanation of how intensive nutritional counseling and a medically tailored meal delivery program can support patients in adjusting to their diagnosis. This leads to active coping through intentional self-care, resulting in perceived positive behavior changes and potentially increasing treatment effectiveness, which is consistent with our results.

Our study found no differences in the PFS and OS between the nutrition counseling and usual care groups. This finding is consistent with that of a systematic review by Kiss et al., which examined nutritional interventions in patients with lung cancer treated with chemotherapy and/or radiotherapy [26], as well as other relevant studies [33]. Evidence-based practice guidelines from 2013 and studies on patients with advanced NSCLC in 2014 also concluded that dietary counseling and/or oral supplements do not enhance survival outcomes, which is consistent with our study’s findings [25,37]. 

Individualized nutritional support has been shown to reduce the risk of mortality and improve functional and quality of life outcomes in different types of cancer patients, including lung cancer patients [16].

### 6.4. QoL Assessment and Others

QoL should be considered a crucial endpoint in clinical trials and is typically included in assessments published in high-impact journals [34]. In the present study, we did not observe a significant difference in QoL from nutritional counseling, which is consistent with the findings of Poulsen et al. [22], and systematic reviews by Kiss et al. [26] and Ueshima J et al. [27]. However, contrary to our results, dietary counseling has been shown to significantly improve the QoL in patients with cancer undergoing chemotherapy, radiation, and a combination of chemotherapy and radiotherapy [14,30,32]. Our study recruited patients with newly diagnosed lung cancer who had undergone first-line treatment. Few comparable studies exist, as most either focused on single treatments, included mixed cancer types, or were conducted on different cancer types altogether. Our study included patients treated with chemotherapy as well as newer treatments, such as targeted therapy and immunotherapy, which typically have fewer adverse events and better clinical outcomes. These treatment differences may have influenced the QoL and nutritional outcomes in our findings. A retrospective French study by Gouez et al. highlighted the fact that severe malnutrition was significantly associated with the efficacy of immunotherapy following chemotherapy in lung cancer, based on a univariate analysis [18]. However, a multivariate analysis did not show an association between nutritional status and treatment efficacy [18]. Furthermore, the study indicated that a weight loss of 1% per month was associated with lower survival rates [18].

In addition to nutritional counseling, dietary supplementation may enhance body weight and nutritional status. The evidence from systematic reviews and meta-analyses [28] suggest that oral nutritional supplementation can increase body weight in patients with cancer, including those with gastrointestinal, head and neck, and lung cancers, receiving chemotherapy. This benefit is particularly notable in populations with a high risk of malnutrition, such as older adult patients, those with a low baseline body weight, women, and non-Asian patients. Furthermore, oral nutritional supplementation has been associated with improvements in PG-SGA scores and a significantly enhanced quality of life [28]. The narrative review indicated that personalized dietary counseling, enhanced protein intake, and omega-3 fatty acid supplementation may positively impact patient outcomes [15].

Other factors that could influence study outcomes include the frequency of dietary counseling, which should be more intensive than it was in our study. Initially conducted at baseline, 3–4 weeks, and 12 weeks, counseling sessions may benefit from increased frequency, such as monthly sessions over a 3-month period, or as requested by patients.

According to Buchan et al. [38] patients with cancer report that using an evidence-based, artificial intelligence-powered virtual dietitian benefits their diet, QoL, and symptom management. This method can help patients and families to adhere to dietary recommendations and reduce barriers due to limited hospital time. However, it needs to be established and compared to traditional dietitian services before being adapted to our patients. 

### 6.5. Implications of the Findings

Our RCT study demonstrated benefits from nutritional counseling in newly diagnosed lung cancer patients receiving standard therapy with chemotherapy or new targeted/immunotherapy. These benefits included improved weight change, nutritional intake, and clinical outcomes, with an enhanced best response in an upper-middle-income country where dietitians were limited in most hospitals. This underscores the importance of providing personalized nutritional counseling or adapting AI to support this program. This research may be applicable to other countries with varying income levels or for different types of cancer.

### 6.6. Strengths

This study was the first RCT comparing the outcomes of dietary counseling provided by nutritionists in patients with newly diagnosed lung cancer who were treated with a standard treatment in an upper-middle-income country. We minimized confounding factors by excluding patients with a history of previous treatment or chemotherapy. In addition, patients with other types of cancer were excluded in order to ensure that this study’s outcomes were specific to lung cancer.

### 6.7. Limitations

There are several limitations to this study. First, the imbalance in baseline characteristics, such as a higher QoL and energy intake in the nutrition counseling group, and a lower percentage of lung metastasis in the same group, may have affected the staging and study outcomes. However, using percentage changes for individual efforts may help reduce this imbalance. Second, the methods for measuring the nutritional status and QoL in our study utilized scores specific to our country, making comparisons with other studies uncertain. While we did not design this study to use the Global Leadership Initiative on Malnutrition (GLIM) criteria, it is worth noting that this criteria could be valuable for future research or clinical practice in terms of diagnosing, assessing, and determining the severity of malnutrition [39]. Third, in cases of advanced chronic kidney disease (CKD) stages IV–V, dietary protein intake may need to be restricted. Similarly, acute renal insufficiency, which can be a complication of certain oncological treatments, should also be taken into account. Fourth, the investigators were responsible for recruitment, randomization, data collection, and data analysis for the outcome assessors and patients were not blind to their treatment allocation. Fifth, despite focusing on lung cancer, this study included treatments such as targeted therapy, immunotherapy, chemotherapy, and combination therapy, which might have influenced the outcomes differently. However, this reflects the current real-world situation in the treatment of lung cancer. Further studies should investigate specific treatments using standardized measurements to provide universal comparisons. 

## 7. Conclusions

Dietary counseling intervention in upper-middle-income countries significantly enhances body weight changes, and energy and protein intake; it also provides clinical benefits through significant improvement of the best response. However, it does not impact other nutritional outcomes or survival in lung cancer patients undergoing definitive treatment. Therefore, timely nutritional intervention can effectively address early malnutrition and should be prioritized for all patients with lung cancer, regardless of their treatment regimen. Nevertheless, dietary counseling alone may not be sufficient to optimize all aspects of nutritional status, and additional dietary supplements may be beneficial. 

## Figures and Tables

**Figure 1 jcm-13-05236-f001:**
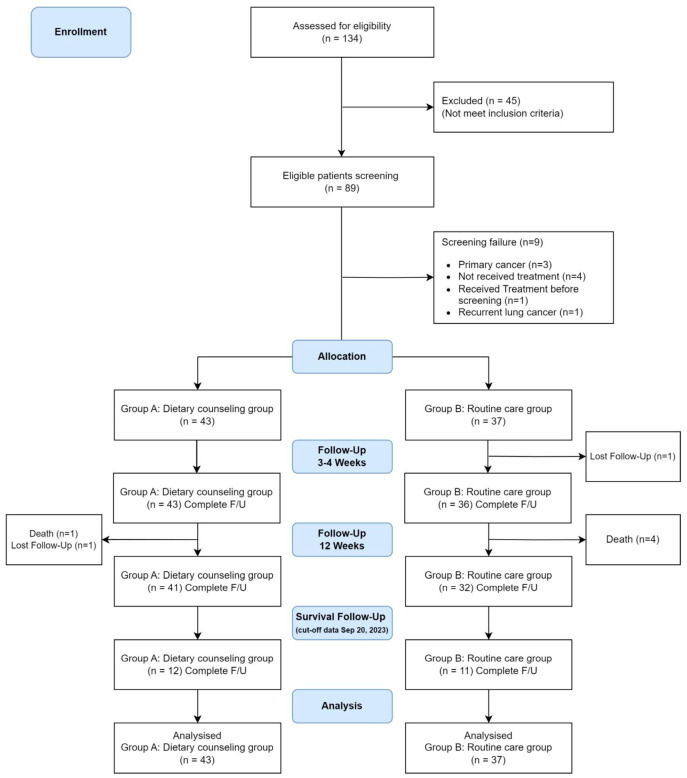
Consort diagram.

**Figure 2 jcm-13-05236-f002:**
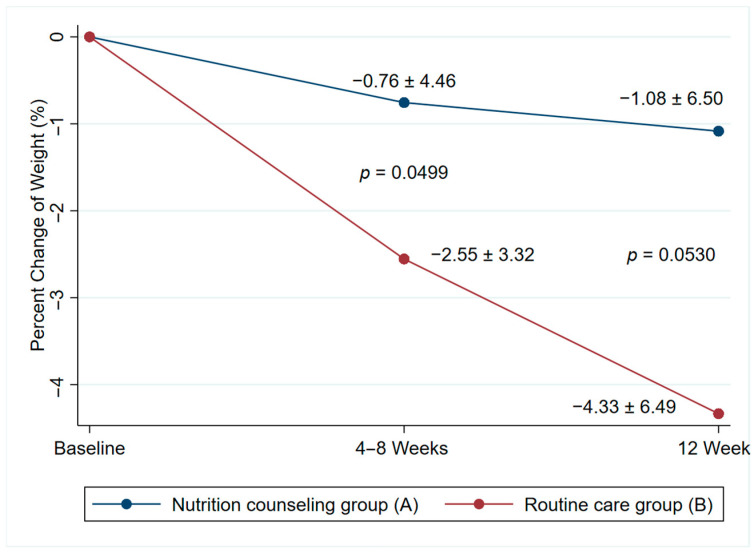
Percentage change of body weight.

**Table 1 jcm-13-05236-t001:** Baseline demographic characteristics.

Characteristics	Total (*n* = 80)	Nutrition Counseling (*n* = 43)	Routine Care (*n* = 37)	*p*-Value
Age, year (mean ± SD)	62.99 ± 10.35	63.56 ± 10.53	62.32 ± 10.25	0.598 *
Sex: no (%)				0.760 ^¶^
• Male	49 (61.25)	27 (62.79)	22 (59.46)	
• Female	31 (38.75)	16 (37.21)	15 (40.54)	
Histopathology: no. (%)				0.518 ^#^
• Adenocarcinoma	59 (73.25)	29 (67.44)	30 (81.08)	
• Squamous-cell carcinoma	14 (17.50)	9 (20.93)	5 (13.51)	
• Small-cell carcinoma	6 (7.50)	4 (9.30)	2 (5.41)	
• Other (poorly differentiated carcinoma)	1 (1.25)	1 (2.33)	0 (0)	
Staging of lung cancer				0.383 ^¶^
• III	21 (26.25)	13 (30.23)	8 (21.62)	
• IV	59 (73.25)	30 (69.77)	29 (78.38)	
Site of metastases: no. (%)				
• Lung	41 (51.25)	17 (39.53)	24 (64.86)	0.024 ^¶^
• Brain	14 (17.50)	6 (13.95)	8 (21.62)	0.394 ^#^
• Bone	13 (16.25)	8 (18.60)	5 (13.51)	0.762 ^#^
• Liver	6 (7.50)	2 (4.65)	4 (10.81)	0.297 ^¶^
• Other	28 (35.00)	14 (32.56)	14 (37.84)	0.622 ^¶^
Comorbid diseases: no. (%)				
• No comorbidity	32 (40.00)	14 (32.59)	18 (48.65)	0.166 ^¶^
• Hypertension	30 (37.50)	18 (41.86)	12 (32.43)	0.385 ^¶^
• Dyslipidemia	25 (31.25)	13 (30.23)	12 (32.43)	0.832 ^¶^
• COPD	11 (13.75)	5 (11.63)	6 (16.22)	0.552 ^¶^
• Diabetes mellitus	10 (12.50)	8 (18.60)	2 (5.41)	0.097 ^#^
• Liver disease	8 (10.00)	5 (11.63)	3 (8.11)	0.719 ^#^
• Gouty arthritis	8 (10.00)	3 (6.98)	5 (13.51)	0.461 ^#^
• Chronic kidney disease	6 (7.50)	1 (2.33)	5 (13.51)	0.090 ^#^
• Others ** (≤5 cases/each condition)	23 (28.75)	12 (27.91)	11 (29.73)	0.857 ^¶^
Smoking status: no. (%)				0.932 ^¶^
• Never smoker	35 (43.75)	19 (44.19)	16 (43.24)	
• Ex-smoker or Current Smoker	45 (56.25)	24 (55.81)	21 (56.76)	
ECOG performance status: no. (%)				0.186 ^#^
• 0	17 (21.25)	12 (27.91)	5 (13.51)	
• 1	54 (67.50)	28 (65.12)	26 (70.27)	
• 2	9 (11.25)	3 (6.98)	6 (16.22)	
Biomarkers: no. (%)				
• EGFR mutation				0.517 ^#^
➢ Positive	25 (31.25)	13 (30.23)	12 (32.43)	
➢ Negative	42 (52.50)	21 (48.84)	21 (56.76)	
➢ Not Done	13 (16.25)	9 (20.93)	4 (10.81)	
• ALK fusion				1.000 ^#^
➢ Positive	6 (7.50)	3 (6.98)	3 (8.11)	
➢ Negative	41 (51.25)	22 (51.16)	19 (51.35)	
➢ Not Done	13 (16.25)	18 (41.86)	15 (40.54)	
• ROS1 fusion				0.851 ^#^
➢ Negative	26 (32.50)	15 (34.88)	11 (29.73)	
➢ Positive	3 (3.75)	2 (4.65)	1 (2.70)	
➢ Not Done	51 (63.75)	26 (60.47)	25 (67.57)	
• PD-L1 TPS				0.596 ^#^
➢ <1%	25 (31.25)	16 (37.21)	9 (24.32)	
➢ 1–49%	6 (7.50)	3 (6.98)	3 (8.11)	
➢ ≥50%	12 (15.00)	5 (11.63)	7 (18.92)	
➢ Not Done	37 (46.25)	19 (44.19)	18 (48.65)	
Treatment (First line): no. (%)				
• Stage III				1.000 ^#^
➢ CCRT or sequential radiotherapy	10 (12.50)	6 (46.15)	4 (50.00)	
➢ Palliative chemotherapy	8 (10.00)	5 (38.46)	3 (37.50)	
➢ Others (chemotherapy + immunotherapy, chemotherapy, surgery)	3 (3.75)	2 (15.38)	1 (12.50)	
• Stage IV				0.127 ^#^
➢ Targeted therapy	20 (25.50)	11 (36.67)	9 (31.03)	
➢ Palliative chemotherapy	26 (32.50)	16 (53.33)	10 (34.48)	
➢ Best supportive care	2 (2.50)	0 (0)	2 (6.90)	
➢ Others	11 (13.75)	3 (10.00)	8 (27.59)	0.182 ^#^
- Chemotherapy + targeted therapy	3 (27.27)	0 (0)	3 (37.50)	
- Chemotherapy + immunotherapy	5 (45.45)	3 (100.00)	2 (25.00)	
- Immunotherapy	3 (27.27)	0 (0)	3 (37.50)	

* *T*-Test. ^¶^ Chi-Squared Test. ^#^ Fisher’s exact test. ** Benign prostate prostatic hyperplasia, Osteoarthritis, Atrial fibrillation, Thalassemia Trait, Glaucoma, Endometrial Hyperplasia, Gall Stone, Hyperuricemia, Cervical spondylosis, Rheumatic heart disease, Aortic aneurysm, Migraine, Compression Fracture, Thalassemia, Renal Calculi, Diverticulitis, Asthma, Cutaneous lupus erythematosus). Abbreviations: COPD—chronic obstructive pulmonary disease; ECOG—Eastern Cooperative Oncology Group; EGFR—epidermal growth factor receptor; ALK—anaplastic lymphoma kinase; ROS1—ROS1 gene; PD-L1—programmed death ligand 1; TPS—tumor proportional score; CCRT—concurrent chemoradiotherapy.

**Table 2 jcm-13-05236-t002:** Baseline nutritional status.

Characteristics	Total (*n* = 80)	Nutrition Counseling (*n* = 43)	Routine Care (*n* = 37)	*p*-Value
Body weight: (mean ± SD)	54.88 ± 10.29	55.41 ± 11.11	54.26 ± 9.36	0.621 *
Percent of weight loss in 1 month: no. (%)				0.055
• <5%	58 (72.50)	35 (81.40)	23 (62.16)	
• ≥5%	22 (27.50)	8 (18.60)	14 (37.84)	
BMI range: no (%)	21.47 ± 3.84	21.76 ± 3.97	21.13 ± 3.70	0.466
• <18.5 (underweight)	17 (21.25)	9 (20.93)	8 (21.62)	0.940
• 18.5–22.9 (normal)	37 (46.25)	17 (39.53)	20 (54.05)	0.194
• ≥23 (overweight)	26 (32.50)	17 (39.53)	9 (24.32)	0.148
Nutrition score (NT-2013 score): mean ± SD	7.93 ± 2.61	7.81 ± 2.58	8.05 ± 2.68	0.684 *
QoL score: (mean ± SD)	46.76 ± 9.27	49.30 ± 7.90	43.81 ± 9.95	0.007 *
Grip strength (kg): (mean ± SD) **	25.53 ± 9.36	24.71 ± 10.34	26.92 ± 7.48	0.446 *
Gait speed (s/4 m): median (P25, P75) ***	4.03 (3.42, 6.56)	4.01 (3.39, 5.25)	4.75 (3.49, 6.58)	0.429 ^#^
Serum albumin (g/dL): (mean ± SD)	3.84 ± 0.44	3.92 ± 0.41	3.76 ± 0.46	0.111 *
Absolute lymphocyte count (cell/mm^3^), (mean ± SD)	1883.47 ± 747.88	1835.47 ± 833.57	1939.26 ± 640.95	0.539 *
Energy intake (kcal/day): (mean ± SD) ****	1390.20 ± 276.66	1447.95 ± 219.12	1316.70 ± 324.86	0.041 *
Protein intake (gm/day): (mean ± SD) ****	50.69 ± 12.69	53.21 ± 12.16	47.48 ± 12.82	0.052 *

* *T*-Test. ^#^ Mann–Whitney-U Test. ** *n* = 46 (A = 29, B = 17). *** *n* = 42 (A = 26, B = 16). **** *n* = 75 (A = 42, B = 33). BMI—body mass index; NT 2013—nutritional assessment; QoL—Quality of life.

**Table 3 jcm-13-05236-t003:** Nutrition Outcomes.

Outcomes	Nutrition Counseling	Routine Care	*p*-Value
Body weight (mean ± SD)			
• Next 3–4 weeks	54.91 ± 10.85	53.38 ± 8.95	0.504 *
• Next 12 weeks	55.28 ± 11.37	53.62 ± 9.39	0.506 *
BMI (mean ± SD)			
• Next 3–4 weeks	21.53 ± 3.80	20.73 ± 3.50	0.343 *
• Next 12 weeks	21.60 ± 3.82	20.80 ± 3.53	0.359 *
Nutrition score (mean ± SD)			
• Next 3–4 weeks	6.46 ± 2.27	7.09 ± 3.00	0.308 *
• Next 12 weeks	5.03 ± 2.25	6.13 ± 2.69	0.067 *
QoL (mean ± SD)			
• Next 3–4 weeks	49.98 ± 9.00	45.58 ± 10.74	0.060 *
• Next 12 weeks	52.31 ± 8.11	49.67 ± 11.00	0.255 *
Grip strength			
• Next 3–4 weeks	25.28 ± 8.94	27.45 ± 10.20	0.529 *
• Next 12 weeks	28.17 ± 10.14	23.01 ± 8.05	0.106 *
Gait speed (mean ± SD)			
• Next 3–4 weeks	4.47 ± 2.76	4.27 ± 1.28	0.814 *
• Next 12 weeks	4.43 ± 1.86	4.04 ± 1.17	0.485 *
Serum albumin (mean ± SD)			
• Next 3–4 weeks	3.84 ± 0.46	3.75 ± 0.47	0.397 *
• Next 12 weeks	4.05 ± 0.43	3.85 ± 0.49	0.077 *
Absolute lymphocyte count (mean ± SD)			
• Next 3–4 weeks	1812.796 ± 671.36	1941.17 ± 804.67	0.445 *
• Next 12 weeks	1806.32 ± 685.87	1910.95 ± 783.53	0.545 *
Energy intake (mean ± SD)			
• Next 3–4 weeks	1508.34 ± 210.23	1368.10 ± 326.31	0.030 *
• Next 12 weeks	1594.19 ± 255.76	1395.02 ± 314.00	0.006 *
Protein intake (mean ± SD)			
• Next 3–4 weeks	56.50 ± 11.61	52.41 ± 14.22	0.183 *
• Next 12 weeks	61.39 ± 12.20	51.03 ± 14.55	0.002 *

* *T*-Test.

**Table 4 jcm-13-05236-t004:** Percentage changes.

	Group A	Group B	*p*-Value
Percentage of weight change (%)			
• Next 3–4 weeks (*n* = 78, A = 43, B = 35)	−0.76 ± 4.46	−2.55 ± 3.32	
Median (P25, P75)	0 (−2.72, 0.75)	−2.244 (−5.33, 0)	0.0499 ^#^
• Next 12 weeks (*n* = 73, A = 41, B = 32)	−1.08 ± 6.50	−4.33 ± 6.49	
Median (P25, P75)	0 (−4.76, 4)	−3.14 (−7.08, 0.12)	0.053 ^#^
Percentage of body mass index (BMI) change (%)			
• Next 3–4 weeks (*n* = 78, A = 43, B = 35)	−0.84 ± 4.48	−2.49 ± 3.87	
Median (P25, P75)	0 (−2.96, 0.75)	−20.4 (−5.33, 0)	0.061 ^#^
• Next 12 weeks (*n* = 73, A = 41, B = 32)	−1.48 ± 6.91	−4.47 ± 6.42	
Median (P25, P75)	−1.98 (−5.25, 4.00)	−31.4 (−7.70, 0)	0.093 ^#^
Percentage of grip strength change (%)			
• Next 3–4 weeks (*n* = 31, A = 19, B = 12)	11.65 ± 29.16	−2.31 ± 15.01	
Median (P25, P75)	3.07 (−0.75, 14.75)	−1.76 (−15.19, 9.79)	0.209 ^#^
• Next 12 weeks (*n* = 27, A = 18, B = 9)	15.20 ± 19.49	−9.09 ± 19.30	
Median (P25, P75)	11.54 (3.75, 27.09)	−5.21 (−18.40, −3.23)	0.006 ^#^
Percentage of gait speed change (%)			
• Next 3–4 weeks (*n* = 26, A = 16, B = 10)	−2.29 ± 25.63	−2.29 ± 25.63	
Median (P25, P75)	3.24 (−20.57, 10.07)	−9.57 (−25.86, 17.00)	0.916 ^#^
• Next 12 weeks (*n* = 26, A = 17, B = 9)	7.62 ± 54.06	−7.55 ± 24.57	
Median (P25, P75)	−4.77 (39.90, 33.97)	−16.00 (−18.08, 4.63)	0.609 ^#^
Percentage of serum albumin change (%)			
• Next 3–4 weeks (*n* = 31, A = 19, B = 12)	−1.43 ± 10.33	0.68 ± 12.95	
Median (P25, P75)	−2.38 (−5.71, 5.00)	0 (−6.67, 5.00)	0.896 ^#^
• Next 12 weeks (*n* = 27, A = 18, B = 9)	3.71 ± 10.69	2.00 ± 16.92	
Median (P25, P75)	2.47 (0, 8.34)	0 (−6.07, 9.41)	0.343 ^#^
Percentage of absolute lymphocyte count change (%)			
• Next 3–4 weeks (*n* = 78, A = 43, B = 35)	71.35 ± 322.47	0.84 ± 29.44	
Median (P25, P75)	−0.86 (−23.25, 15.87)	−0.29 (−20.85, 27.49)	0.972 ^#^
• Next 12 weeks (*n* = 73, A = 41, B = 32)	78.14 ± 362.91	−3.04 ± 30.35	
Median (P25, P75)	−3.06 (−33.87, 19.73)	−4.13 (−23.23, 19.44)	0.991 ^#^
Percentage of energy intake change (%)			
• Next 3–4 weeks (*n* = 72, A = 41, B = 31)	5.67 ± 13.92	2.27 ± 16.99	
Median (P25, P75)	6.67 (−3.91, 15.24)	4.69 (−10.00, 11.11)	0.197 ^#^
• Next 12 weeks (*n* = 67, A = 37, B = 30)	13.37 ± 21.61	4.91 ± 18.02	
Median (P25, P75)	13.70 (1.82, 23.81)	3.35 (−7.69, 15.11)	0.065 ^#^
Percentage of protein intake change (%)			
• Next 3–4 weeks (*n* = 72, A = 41, B = 31)	9.06 ± 24.18	10.51 ± 26.75	
Median (P25, P75)	8.33 (−6.78, 25.00)	7.50 (−9.09, 23.81)	0.923
• Next 12 weeks (*n* = 67, A = 37, B = 30)	18.82 ± 30.30	6.13 ± 19.87	
Median (P25, P75)	15.45 (−3.84, 40.00)	0 (−4.50, 20.00)	0.108
Percentage of QoL change (%)			
• Next 3–4 weeks (*n* = 73, A = 40, B = 33)	2.49 ± 19.04	6.89 ± 21.84	
Median (P25, P75)	1.96 (−5.88, 12.77)	6.38 (−4.55, 19.35)	0.344
• Next 12 weeks (*n* = 68, A = 38, B = 30)	9.98 ± 28.07	18.03 ± 28.47	
Median (P25, P75)	5.66 (−10.87, 28.89)	18.65 (1.85, 38.46)	0.221
Percentage of nutrition score change (%)			
• Next 3–4 weeks (*n* = 75, A = 41, B = 34)	−11.28 ± 36.15	−7.26 ± 39.61	
Median (P25, P75)	−18.18 (−33.33, 0)	−12.70 (−33.33, 9.09)	0.564
• Next 12 weeks (*n* = 69, A = 39, B = 30)	−26.01 ± 47.99	−14.54 ± 44.83	
Median (P25, P75)	−44.44 (−58.33, 0)	−22.50 (−40.00, 0)	0.099

^#^ Mann–Whitney-U Test.

**Table 5 jcm-13-05236-t005:** Efficacy of first-line treatment.

	Nutrition Counseling	Routine Care	*p*-Value
Best response of first-line treatment			0.010
• Partial response	26 (60.47)	18 (48.65)	
• Stable disease	16 (37.21)	10 (27.03)	
• Progressive disease	0 (0)	7 (18.92)	
• Not evaluate	1 (2.33)	2 (5.41)	
Median progression-free survival (month)	9.54 (6.46, 14.36)	9.90 (4.33, 24.20)	0.473 ^#^
Median overall survival (month)	18.82 (12.92, 32.89)	17.18 (7.31, 31.21)	0.516 ^#^

^#^ Log-rank test.

**Table 6 jcm-13-05236-t006:** Subsequent Treatments.

	Nutrition Counseling	Routine	*p*-Value
Treatment (Second line)			0.103 ^#^
• Chemotherapy	7 (30.43)	11 (61.11)	
• Targeted therapy	6 (26.09)	5 (27.78)	
• Chemotherapy in combination with other treatment(s)	4 (17.39)	0 (0)	
• Immunotherapy	6 (26.09)	2 (11.11)	
Treatment (Third line)			0.273 ^#^
• Chemotherapy	5 (62.50)	4 (66.67)	
• Targeted therapy	33.33	0 (0)	
• Chemotherapy in combination with other treatment(s)	2 (25.00)	0 (0)	
• Immunotherapy	1 (12.50)	0 (0)	
Treatment (Fourth line)			0.333 ^#^
• Chemotherapy	0 (0)	2 (100.00)	
• Chemotherapy in combination with other treatment(s)	1 (100.00)	0 (0)	
Total Line (Systemic therapy)			0.394 ^#^
• 0	0 (0)	3 (8.11)	
• 1	20 (46.51)	16 (43.24)	
• 2	15 (34.88)	12 (32.43)	
• 3	7 (16.28)	4 (10.81)	
• 4	1 (2.33)	2 (5.41)	

^#^ Fisher’s exact test.

## Data Availability

Data are available upon reasonable request. De-identified data are currently held by the Medical Oncology Unit at the Faculty of Medicine at Chiang Mai University, in Thailand.

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
