# Peer review of "Dietary Counseling Outcomes in Patients with Lung Cancer in an Upper-Middle-Income Country: An Open-Label Randomized Controlled Trial"

_jcm, 2024, doi:10.3390/jcm13175236_

Round 1

Reviewer 1 Report

Comments and Suggestions for Authors

I would like to commend the authors for this study, as I believe the topic is highly relevant and of significant interest. In particular, nutrition in cancer patients is sometimes overlooked, and dietary consultation is of primary importance. To improve the quality of the manuscript, I have a few minor comments:

1. In the description of associated comorbidities, acute or chronic renal insufficiency was not mentioned. In cases of advanced chronic kidney disease (CKD) stages IV-V, this could limit protein intake through diet. Similarly, acute renal insufficiency, a possible complication of certain oncological treatments, should also be considered. I believe these aspects should be mentioned in the study's limitations.

2. Did the nutritional counseling include the administration of oral nutritional supplements (ONS)? While I understand it may be challenging to determine due to the heterogeneity of nutritional formulations and the sample, it should be indicated more clearly.

3. An assessment based on GLIM criteria to diagnose malnutrition at the start of the study would have been interesting.

4. Some citations are quite old. Updating the references with more recent references, where possible, could improve the quality of the manuscript.

Author Response

Dietary Counseling Outcomes in Patients with Lung Cancer in an Upper Middle-Income Country: An Open-Label Randomized Controlled Trial

Reviewer 1.

Thank you very much for taking the time to review this manuscript and for providing such useful recommendations. Please find the detailed responses below and the corresponding revisions/corrections highlighted/in track changes in the re-submitted files

Comment 1:  In the description of associated comorbidities, acute or chronic renal insufficiency was not mentioned. In cases of advanced chronic kidney disease (CKD) stages IV-V, this could limit protein intake through diet. Similarly, acute renal insufficiency, a possible complication of certain oncological treatments, should also be considered. I believe these aspects should be mentioned in the study's limitations.

Response 1:  I have added the comorbidities, including chronic kidney disease (1 in the nutrition counseling group and 5 in the routine care group), noting that all CKD patients were at stage III. I have also included your suggestion in the study's limitations: “in cases of advanced chronic kidney disease (CKD) stages IV-V, dietary protein intake may need to be restricted. Similarly, acute renal insufficiency, which can be a complication of certain oncological treatments, should also be taken into account."

Comment 2: Did the nutritional counseling include the administration of oral nutritional supplements (ONS)? While I understand it may be challenging to determine due to the heterogeneity of nutritional formulations and the sample, it should be indicated more clearly.

Response 2:  The nutritionist advised patients to meet their energy and protein requirements based on the ESPEN guidelines, considering individual comorbidities such as diabetes, CKD, and others. For patients unable to meet their nutritional needs through normal food intake, oral nutritional supplements (ONS) were recommended. However, due to the high cost of ONS in an upper-middle-income country, only some patients were able to access them. Patients recorded the amount and type of ONS they consumed, which were reviewed during interviews with the nutritionist. The dietitian then assessed the nutrition scores, as well as the details of energy and protein intake, based on these interviews. The results are presented in Tables 2, 3, and 4, and the nutritional intervention section has also been revised.

Comment 3: An assessment based on GLIM criteria to diagnose malnutrition at the start of the study would have been interesting.

Response 3: While we did not design this study to use the GLIM criteria, it is worth noting that these criteria could be valuable for future research or clinical practice in terms of diagnosing, assessing, and determining the severity of malnutrition. We have included this consideration in the limitations section of our study.

Comment 4: Some citations are quite old. Updating the references with more recent references, where possible, could improve the quality of the manuscript.

Response 4:  We would like to express our gratitude for your valuable suggestions. We update the references with more recent references.

Reviewer 2 Report

Comments and Suggestions for Authors

This is a well-conducted study with the potential to make a significant contribution to the field of lung cancer care. I recommend publication with minor revisions to address the following specific areas:

1. Randomization and Blinding:

  • Please provide more details on the randomization method used, including the block size and allocation ratio.
  • Clearly state whether the investigators, outcome assessors, and patients were blinded to treatment allocation. If not, acknowledge this as a limitation.

2. Timing of Baseline Assessments:

  • Please specify when the baseline measurements were conducted relative to the start of the intervention.

3. Presentation of Primary Outcome Data:

  • In addition to the mean percentage change, please provide the actual mean values for each group at baseline and 12 weeks to facilitate a better understanding of the changes.

4. Reporting of Secondary Outcomes:

  • Report the specific p-values and effect sizes for the secondary outcomes, even if they were not statistically significant. This will provide a more complete picture of the results.

5. Adverse Event Reporting:

  • Define what constitutes a "significant" adverse event to clarify the reporting criteria.

6. Language Refinement:

  • Please review the manuscript for grammatical errors, syntax issues, and typos.

By addressing these points, the manuscript will be strengthened and more effectively communicate the study's findings.

Comments on the Quality of English Language

Moderate editing of English language required.

Author Response

Dietary Counseling Outcomes in Patients with Lung Cancer in an Upper Middle-Income Country: An Open-Label Randomized Controlled Trial

Reviewer 2

Comment 1: Randomization and Blinding:

  • Please provide more details on the randomization method used, including the block size and allocation ratio.
  • Clearly state whether the investigators, outcome assessors, and patients were blinded to treatment allocation. If not, acknowledge this as a limitation.

Response 1: The patients were randomly allocated using computer-generated random number codes with blocked randomization (block size of four). The sequence was not concealed or blinded until the interventions were assigned. The investigators were responsible for recruitment, randomization, data collection, and data analysis for the outcome assessors, which was already acknowledged as a limitation.

Comment 2: Timing of Baseline Assessments:

  • Please specify when the baseline measurements were conducted relative to the start of the intervention.

Response 2: The baseline measurements were conducted within three days before the intervention.

Comment 3: Presentation of Primary Outcome Data:

  • In addition to the mean percentage change, please provide the actual mean values for each group at baseline and 12 weeks to facilitate a better understanding of the changes.

Response 3: The actual mean values of body weight for each group at baseline and at 12 weeks are shown in Tables 2 and 3, respectively.

Comment 4: Reporting of Secondary Outcomes:

  • Report the specific p-values and effect sizes for the secondary outcomes, even if they were not statistically significant. This will provide a more complete picture of the results.

Response 4: The report of secondary outcomes and p-values is shown in Tables 4 and 5.

Comment 5: Adverse Event Reporting:

  • Define what constitutes a "significant" adverse event to clarify the reporting criteria.

Response 5: A significant serious adverse event refers to an adverse event that leads to death, hospitalization, disability or permanent damage, or a congenital anomaly/birth defect.

Comment 6: Language Refinement:

  • Please review the manuscript for grammatical errors, syntax issues, and typos. By addressing these points, the manuscript will be strengthened and more effectively communicate the study's findings.

Response 6: We already used MDPI Author Services for assistance with grammar, syntax, and typographical issues, and we will resend it for review after the corrections are made.